# Hydrogen Peroxide Increases during Endodormancy and Decreases during Budbreak in Grapevine (*Vitis vinifera* L.) Buds

**DOI:** 10.3390/antiox10060873

**Published:** 2021-05-29

**Authors:** Francisco Javier Pérez, Ximena Noriega, Sebastián Rubio

**Affiliations:** Laboratorio de Bioquimica Vegetal, Facultad de Ciencias, Universidad de Chile, Casilla 653, Las Palmeras, Ñuñoa 3425, Chile; XNoriegaguerrero@gmail.com (X.N.); rubio604@hotmail.com (S.R.)

**Keywords:** auxin, cytokinin, budbreak, dormancy, grapevine buds, hydrogen peroxide, peroxidases

## Abstract

Changes in the level of hydrogen peroxide (H_2_O_2_) is a good indicator to monitor fluctuations in cellular metabolism and in the stress responses. In this study, the changes in H_2_O_2_ content during bud endodormancy (ED) and budbreak were analysed in grapevine (*Vitis vinifera* L.). The results showed a gradual increase in the H_2_O_2_ content during the development of bud ED, which was mainly due to an increase in the activity of peroxidases (PODs). The maximum H_2_O_2_ content reached in the grapevine buds coincided with the maximum depth of bud ED. In contrast, during budbreak, the H_2_O_2_ content decreased. As the plant hormones cytokinin (CK) and auxin play an important role in budbreak and growth resumption in grapevine, the effect of exogenous applications of H_2_O_2_ on the expression of genes involved in CK and auxin metabolism was analysed. The results showed that H_2_O_2_ represses the expression of the CK biosynthesis genes *VvIPT3a* and *VvLOG1* and induces the expression of the CK-inactivating gene *VvCKX3*, thus reducing potentially the CK content in the grapevine bud. On the other hand, H_2_O_2_ induced the expression of the auxin biosynthesis genes *VvAMI1* and *VvYUC3* and of the auxin transporter gene *VvPIN3*, thus increasing potentially the auxin content and auxin transport in grapevine buds. In general, the results suggest that H_2_O_2_ in grapevine buds is associated with the depth of ED and negatively regulates its budbreak.

## 1. Introduction

The buds of the grapevine (*Vitis vinifera* L.), similar to the buds of other deciduous fruit trees enter a state of winter recess or endodormancy (ED) to survive winter conditions. In the grapevine bud, ED is induced by the shortening of the photoperiod [1,2,3] and is regulated by endogenous factors that inhibit its growth and development [4]. The plant hormone abscisic acid (ABA) accumulates in the buds of grapevines throughout ED [5,6] and plays a key role in its maintenance and release [7,8,9]. On the other hand, hydrogen peroxide (H_2_O_2_), which is continuously generated from various sources during normal metabolism and is a signalling molecule that mediates responses to biotic and abiotic stresses [10], also accumulates in the buds of grapevine during ED [11,12,13]. Further, H_2_O_2_ has been suggested to play a pivotal role in the control of ED and budbreak [14]. H_2_O_2_ can also be generated by specific enzymes such as the respiratory burst oxidase homologs (RBOH), xanthine oxidase, amine oxidase and cell wall peroxidase [10]. Generally, the plant antioxidant system removes H_2_O_2_ efficiently, and thus the oxidative effects of many stimuli could be mediated via a reduction in the activities of antioxidant enzymes, rather than by increased H_2_O_2_ generation [10]. In grapevine buds, hypoxia, as well as mitochondrial respiration inhibitors, such as potassium cyanide (KCN) and sodium nitroprusside (SNP), increase H_2_O_2_ levels [15]. In *Arabidopsis thaliana*, hypoxia was found to activate Rop-GTPase signalling, which in turn activated RBOH [16]. In grapevines, seven *VvRBOH* genes have been identified and characterised [17], and three of them are expressed in the buds [18]. Peroxidase class III (POD) represents a class of ubiquitous enzymes widely distributed in plants whose primary function is to oxidize molecules at the expense of H_2_O_2_. POD is therefore generally considered an H_2_O_2_ detoxifying enzyme; however, POD can also produce H_2_O_2_. In grapevines and peach, it has been reported that POD can generate H_2_O_2_ through the oxidation of NADH, and this reaction is catalysed by p-coumaric acid [11,19]. Recently, a subfamily of 47 *VvPOD* genes was identified in the grapevine genome [20] and 30 subfamily members were expressed in the bud [18]. Oxidative stress and reactive oxygen species (ROS) have been suggested to play a central role in the release of buds from ED in grapevines [14]. Interestingly, most studies using chemical and physical dormancy breakdown stimuli such as hydrogen cyanamide (HC), sodium azide (AZ) and heat shock (HS) [11,12,13] show a transient increase in H_2_O_2_ levels within the grapevine bud, which has been interpreted as a positive signal for the release of buds from the ED [13,14].

In this study, the changes in H_2_O_2_ levels though ED and budbreak were analysed in grapevine buds. Additionally, the effects of exogenous applications of H_2_O_2_ on the expression of cytokinin (CK) and auxin-related genes in grapevine buds were examined. From the results, we found that the H_2_O_2_ level is associated with the degree of ED and is a negative regulator of budbreak in grapevine buds.

## 2. Materials and Methods

### 2.1. Plant Material

Plant material was collected from 8-year-old (*Vitis vinifera* L. cv. Thompson Seedless) vineyards growing at the experimental station of the Chilean National Institute of Agriculture Research (INIA) located in Santiago, Chile (33◦34′ S). The grapevine plants were watered by immersion and trained in an aerial training system. Six whole canes were randomly collected every 3 weeks starting at the beginning of January (early summer) and ending at the end of August (early spring). The collected canes were cut at both ends, leaving a central section of 10–12 buds for experimental use.

### 2.2. Dormancy Depth

The budbreak response of single-bud cuttings under forced conditions is widely used to describe the depth of dormancy in grapevine [21,22]. Thirty cuttings were mounted on polypropylene sheets and placed in a container with water and transferred to a growth chamber set at 23 ± 2 °C with 16 h (h) light. The budbreak was assessed every 5 days (d) during a period of 30 d, the appearance of a green tip was the signal that indicated the beginning of the budbreak. This procedure was repeated for the different collection dates. The time required for each sample to reach budbreak, including right-censored observations of the buds that did not break during the treatment, was adjusted to the survival distribution function by the nonparametric Kaplan–Meier method [23]. The first sample of the year collected in early January before the onset of ED [24,25] was used as a reference of the behaviour of budbreak, when growth was not restricted within the buds (paradormant buds). A log-rank test was performed to compare the estimated survival distributions of the reference sample with samples collected at other dates. The greater the difference in the chi-square between the analysed sample and the reference sample, the greater the degree of ED [23].

### 2.3. Chemical Treatments

Single-node cuttings collected on June 11 were painted with 2.5% (*w*/*v*) hydrogen cyanamide (HC) (Dormex, SKW, Trotsberg, Germany), 2% (*w*/*v*) 3-amino-1,2,4-triazole (Sigma, Burlington, VT, USA) or water, as a control. Cuttings were mounted on a polypropylene sheet and placed in a container with water and transferred to the growth chamber set at 23 ± 2 °C under 14 h light forced conditions (FC); samples were removed at each specific point. Experiments inducing the breakage of ED were carried out with buds collected on July 18 and treated with 2.5% (*w*/*v*) HC, 2% (*w*/*v*) 3-amino-1,2,4-triazole or water as a control. After treatments, cuttings were mounted as described above and settled in the growth chamber under FC and the breakage of buds was assessed every 2 d.

### 2.4. H_2_O_2_ Measurements

The H_2_O_2_ concentration was measured by chemiluminescence (CL) based on a cobalt-catalysed oxidation of luminol (5-amino-2,3-dihydro-1,4-phtalazinedione) [26]. Three different buds were analysed for each sampling date and values corresponded to the average of the three samples.

### 2.5. Peroxidase and Catalase Activity

Buds were ground in liquid nitrogen, and the resulting powder was extracted with a buffer containing 0.5 M Tris-HCl, 5 mM DTT, 1 mM MgCl_2_, 10 µM PMSF, 2% insoluble PVP and 12.5% glycerol (pH 7.5). Peroxidase (donor: hydrogen peroxide oxidoreductase EC1.11.1.7) activity was assayed according to [27] by measuring the H_2_O_2_-dependent oxidation of o-phenylendiamine (o-PDA) via spectrophotometry at 450 nanometers (nm) in a mixture with pH 4.5 containing 0. 1 M sodium citrate, 44 mM o-PDA, 1 mM H_2_O_2_ and 5–10 µL of the extract. Catalase activity was determined following O_2_ evolution using a Clark-type oxygen electrode (Hansatech, UK) as according to the methodology of [12].

### 2.6. RNA Purification, and cDNA Synthesis

Total RNA was isolated and purified from dormant buds (0.5 g FW) of Thompson seedless. Total RNA was extracted and purified using the method of Chang et al. [28] modified according to Noriega et al. [29]. The DNA was removed from the sample with RNAse-free DNase I (1 U/µg) (Thermo Scientific, Bedford, MA, USA) at 37 °C for 30 min. The first cDNA strand was synthesised from 1.0 µg of purified RNA with 1 µL of oligo (dT) 12–18 (0.5 µg × µL^−1^) as a primer, 1 µL of dNTP mixture (10 mM) and Superscript® II RT (Invitrogen, CA, USA).

### 2.7. Gene Expression Analysis

The expression analysis of the genes was performed by quantitative real-time PCR (RT-qPCR) using an Eco Real-Time PCR system (Illumina, Inc. San Diego, CA, USA) and KAPA SYBR FAST (KK 4602) qPCR Master Mix (2×). Sequences of grapevine genes were obtained from the grape genomic database (www.genoscope.fr, accessed on 24 May 2021). Primers for amplification were designed using PRIMER3 software [30]. The cDNA was amplified under the following conditions: denaturation at 94 °C for 2 min and 40 cycles of 94 °C for 30 s, 55 °C for 30 s and 72 °C for 45 s. Relative changes in gene expression levels were determined using the 2^−ΔΔCT^ method [31]. Each reaction was performed with at least two biological replicates, each with three technical replicates. *VvUBIQUITIN* (GSVIVT01038617001) and *VvACTIN* (GSVIVT01026580001) were used as reference genes for normalisation.

### 2.8. Data Analysis

Survival analysis, or the time to event analysis, is a category of statistical methods designed specifically to handle a response variable that measures the elapsed time until a specific event occurs (here, budbreak), which may be censored [32]. Kaplan–Meier (KM) survival curves are the simplest way to estimate survival over time when data are censored [23]. The percentage or probability of absence of budbreak is calculated as a function of time after sampling according to the KM method. However, in this study, we used the complement of the probability of the absence of budbreak, which corresponds to the probability of budbreak [23]. A log-rank test was carried out to compare the estimated survival distribution of the reference sample against the other samples. A significant difference indicates differences in the budbreak distribution curves, and the larger the value of the chi-square (χ^2^) is, the greater the difference between the reference and the other sample, thus the greater the ED depth of the sample.

## 3. Results

### 3.1. Depth of Endodormancy and H_2_O_2_ Content in Grapevine Buds

The survival analysis of the budbreak of grapevine buds under forced conditions was carried out on the different collection dates using the nonparametric Kaplan–Meier method [23]. The comparison of the KM probabilistic function between the buds collected before (reference) and after the onset of the ED was performed using a log-rank test. The greater the chi-square between the reference and the samples, the greater the degree of ED [23]. Using this methodology, two phases were distinguished during ED in Thompson seedless grapevines grown in Santiago, Chile. The first phase began in mid-January and lasted until mid-April, and the second phase peaked in late May and ended in late August. During the second phase of the ED, a sustained increase in the H_2_O_2_ content was observed, whose maximum coincided with the maximum depth of the ED (Figure 1). This dormancy pattern is consistent with the transcriptome changes observed during bud development in grapevines [18].

### 3.2. Increases in Peroxidase Activity and H_2_O_2_ Content Coincided through Endodormancy in the Grapevine Buds

In grapevine, it has been reported that peroxidases (PODs) can generate H_2_O_2_ by oxidation of NADH, and this reaction is catalysed by p-coumaric acid [11]. Here, we analysed the POD activity throughout the ED period in Thompson seedless grapevine buds, and this activity was compared with the evolution of the H_2_O_2_ content in the buds (Figure 2a). The results showed that the maximum POD activity coincided with the maximum H_2_O_2_ content, suggesting that the POD activity may be at least partially responsible for the increases in the H_2_O_2_ content in grapevine buds. The increase in POD activity is probably due to an increase in protein abundance, since the expression of *VvPODs* genes increased (see Section 3.3).

### 3.3. Expression Profile of VvPOD Genes throughout the Endodormancy in Grapevine Buds

A total of 47 *VvPOD* genes have been identified in the *V. vinifera* genome [20], and a phylogenetic tree revealed that *VvPODs* showed a relatively close genetic relationship with similar genes from Arabidopsis [20]. Of this large number of *VvPODs*, 30 are expressed in the bud [18], and nine change their expression level during the development of ED. The transcript levels of *VvPODs* taken from microarray data on the development of Tempranillo grape buds grown in the Northern Hemisphere [18] were plotted throughout the bud growth period. The results showed that when the data were adapted to the conditions of the Southern Hemisphere (Figure 2b), only the expression profile of *VvPOD38* was consistent with the increase in POD activity during the development of ED, and *VvPOD25* was expressed slightly during ED but increased before budbreak (Figure 2b).

### 3.4. Expression Profile of RBOH throughout the Endodormancy Period in Grapevine Buds

As one of the major sources of ROS in plants is the NADPH oxidase-catalysed conversion of oxygen to O_2_^−^, the expression of the *RBOH* genes in grapevine buds was analysed. Seven *VvRBOH* genes have been identified in the *V. vinifera* genome [17], and three of them are expressed in the bud [18]. The transcript levels of *VvRBOHs* taken from microarray data on the development of Tempranillo grape buds grown in the Northern Hemisphere [18] were plotted throughout the ED period. The results showed that when the data were adapted to the conditions of the Southern Hemisphere (Figure 3), the expression of *VvRBOHE* was the only family member to have its transcriptional activity significantly altered during the development of ED; however, its maximum expression level did not match the maximum level of H_2_O_2_.

### 3.5. Scavenging Activity of H_2_O_2_ during Endodormancy in Grapevine Buds

It is well known that the ROS scavenging system is involved in the control of H_2_O_2_ in many plant developmental processes [33]. Here, we analysed the expression profile of the catalase genes *VvCAT1* and *VvCAT2* and catalase activity throughout the ED period in grapevine buds. The transcript levels of *VvCAT1* and *VvCAT2* taken from microarray data on the development of Tempranillo grape buds grown in the Northern Hemisphere [18] were plotted throughout the ED period. The results showed that when the data were adapted to the conditions of the Southern Hemisphere (Figure 4b), no major variations in the expression of *VvCAT1* and *VvCAT2* in the buds during their ED period were observed. Catalase activity also did not vary significantly during the ED period in Thompson seedless grapevines, but a peak was detected at the end of July (Figure 4a). As there is a correspondence between catalase activity and expression of catalase genes, the changes in activity reflect the changes in protein abundance.

On the other hand, aminotriazole, an inhibitor of catalase activity [34], rapidly increased H_2_O_2_ content in grapevine buds collected at the end of the ED, and this high H_2_O_2_ level was maintained for longer than when buds were treated with HC (Figure 5a). Furthermore, aminotriazole delayed the sprouting of grapevine buds relative to control buds (Figure 5b).

### 3.6. Variations in the H_2_O_2_ Content during Budbreak in Grapevine Buds under Forced Conditions

To evaluate the changes in the H_2_O_2_ content during forcing budbreak, the H_2_O_2_ content was measured several days after the onset of the treatment in grapevine buds collected in early June, when they were endodormant [24] (Figure 6a). The H_2_O_2_ was also measured a few hours after the onset of the treatment in grapevine buds collected at the end of July, once the ED was finished and the buds were quiescent [24] (Figure 6b). The H_2_O_2_ content decreased rapidly in the quiescent buds (Figure 5b), while in the dormant buds the decrease in the H_2_O_2_ content was much slower (Figure 5a).

### 3.7. Cytokinin-Inactivating Genes and Auxin Biosynthesis Genes Are Induced by H_2_O_2_ in Grapevine Buds, While Cytokinin Biosynthesis Genes Are Repressed

Cytokinin has an antagonistic function to auxin in different developmental processes [35]. Here, we analysed the effect of exogenous applications of H_2_O_2_ on the expression of the CK biosynthesis genes, *VvIPT3a* and *VvLOG1*, and the CK inactivating gene *VvCKX3* in grapevine buds. We selected these genes of CK metabolism because their expression varied greatly during the sprouting of grapevine buds under forced conditions [36]. Additionally, the effect of H_2_O_2_ on the expression of the auxin biosynthesis genes, *VvAMI1* and *VvYUC3*, and the auxin transport gene *VvPIN3* was also analysed for the same reason mentioned above. The results showed that after 48 h of H_2_O_2_ treatment, the expression levels of the CK biosynthesis genes, *VvIPT3a* and *VvLOG1* decreased, while that of the CK-inactivating gene *VvCKX3* increased (Figure 7a). These results suggest that H_2_O_2_ decreases the CK content in the grapevine buds. On the other hand, H_2_O_2_ increased the expression levels of the auxin biosynthesis genes *VvAMI1* and *VvYUC3* and of the auxin transport gene *VvPIN3* (Figure 7b) in grapevine buds. These results suggest that the auxin content and auxin transport are increased in grapevine buds after H_2_O_2_ treatment.

## 4. Discussion

### 4.1. ABA and H_2_O_2_ Accumulate in Grapevine Buds during ED

Under stress conditions, abscisic acid (ABA) and hydrogen peroxide (H_2_O_2_) accumulate in many plant biological systems [37,38]. In grapevine buds, both molecules accumulate during ED [5,6]. The ABA peak occurred before that of H_2_O_2_ [6] and the peak of H_2_O_2_ coincided with the maximum depth of the ED (Figure 1), which suggests that ABA could control the level of H_2_O_2_, which in turn, would be related to the depth of the ED in grapevine buds. In guard cells, ABA induces H_2_O_2_ production presumably via the activity of NADPH-oxidase [39]. In rice roots, ABA induces increases in H_2_O_2_ via activation of cell wall peroxidases [40]. In rice leaves under water stress, ABA controls H_2_O_2_ accumulation through the induction of the catalase gene *OsCATB* [41], and grafting-induced ABA accumulation in cucumber leaves triggers H_2_O_2_ production and enhances the activities of antioxidant enzymes [42]. In grapevine, during bud ED, H_2_O_2_ levels correlated with increases in POD activity and *VvPOD38* gene expression, but not with increases in *VvRBOH* and *VvCAT* gene expression, which suggested that during bud ED, the H_2_O_2_ content is controlled mainly by POD activity. Interestingly, ABA has been reported to regulate the expression of several *VvPOD* genes in grapevine berry skins (Appendix A) including *VvPOD38* [43,44], but its effect on H_2_O_2_ levels has not been described in grapevines. However, the role of ABA in bud ED and budbreak in grapevines has been extensively reported in the literature [7,8,9]. Thus, it has been shown that ABA represses the expression of cell cycle genes in the bud meristem of grapevine [8], and its removal is a key step for dormancy release [7,9]. Additionally, ABA promotes starch synthesis and storage metabolism in endodormant grapevine buds, thus contributing to the establishment of bud ED [6]. On the other hand, although H_2_O_2_ accumulates in grapevine buds during ED, its role in dormancy regulation has been associated mainly with the release of buds from ED rather than with their establishment and maintenance.

### 4.2. The Role of H_2_O_2_ during Bud ED Release and Budbreak

As the application of chemical and physical stimuli that break bud dormancy, such as hydrogen cyanamide (HC), sodium azide (AZ), heat shock (HS) and hypoxia, induces a transient increase in H_2_O_2_ levels [15,45,46], the release of buds from ED has been associated with oxidative and respiratory stress [14]. Recently, it has been demonstrated that HC transiently increases H_2_O_2_ levels and the expression of the *VvPOD72* (GSVIVT01029771001) gene in paradormant grapevine buds [45], and in a proteomic study, it was demonstrated that HC induces the expression of *VvPOD25* [19], a gene that is weakly expressed during ED and more highly expressed during budbreak. All these studies suggest that the transient increase in H_2_O_2_ levels observed in grapevine buds after being treated with HC is due either to an increase in the expression of *VvPOD* genes or to the inhibition of catalase activity [12,13]. The transient increase in H_2_O_2_ levels could act as a secondary messenger, triggering the expression of genes related to endodormancy release [13]. However, when H_2_O_2_ levels remained high for a long period of time, such as after aminotriazole treatment, the budbreak was delayed (Figure 7). In rose flower buds, the content of H_2_O_2_ decreases during the outgrowth process as a result of the activation of the ascorbic acid-glutathione (ASA-GSH) cycle [47]. Additionally, exogenous applications of H_2_O_2_ to rose buds delay their outgrowth and inhibit the expression of key genes involved in bud outgrowth, such as *VACUOLAR INVERTASE* (*RhVI*), *EXPANSIN* (*RhEXP*), *PROLIFERATING CELL NUCLEAR ANTIGEN* (*RhPCNA*) and *CYCLIN D3* (*RHCYCD3*) [47]. These results indicate that H_2_O_2_ negatively controls the outgrowth of rose buds. Our results on the budbreak of quiescent grapevine buds under forced conditions showed a rapid drop in the H_2_O_2_ content with budbreak progress, while in dormant buds, this drop was slower, indicating that H_2_O_2_ should be removed from the bud by the antioxidative system before budbreak and the resumption of growth begins. In addition, the fact that exogenous applications of H_2_O_2_ reduce the CK content by inducing the expression of the CK degrading gene (*VvCKX3*) and repressing the expression of the CK biosynthesis genes (*VvIPT3a* and *VvLOG1*) suggests that H_2_O_2_ negatively affects budbreak, since CK positively modulates cell division and proliferation in meristematic tissue [48]. One of the first cellular events that occurs before the onset of budbreak in grapevine buds is the increase in the expression of CK biosynthesis genes as well as for genes related to the cell cycle [49,50]. In tomato plants, H_2_O_2_ promotes auxin biosynthesis in the apex, which, in turn, inhibits CK biosynthesis and subsequent bud outgrowth [51]. This result agrees with our results obtained in the grapevine buds which indicates an increase in the expression of auxin biosynthesis and transport genes after H_2_O_2_ treatments (Figure 6b).

## 5. Conclusions

During the development of ED in grapevine buds, the level of H_2_O_2_ increased as a result of a higher expression of the *VvPODs* genes, whose transcription could be regulated by ABA. On the contrary, during budbreak, the H_2_O_2_ level decreased. This difference, in changes in the level of H_2_O_2_ during bud development, is consistent with the transcriptional effect of H_2_O_2_ on the genes that regulate CK and auxin metabolism.

## Figures and Tables

**Figure 1 antioxidants-10-00873-f001:**
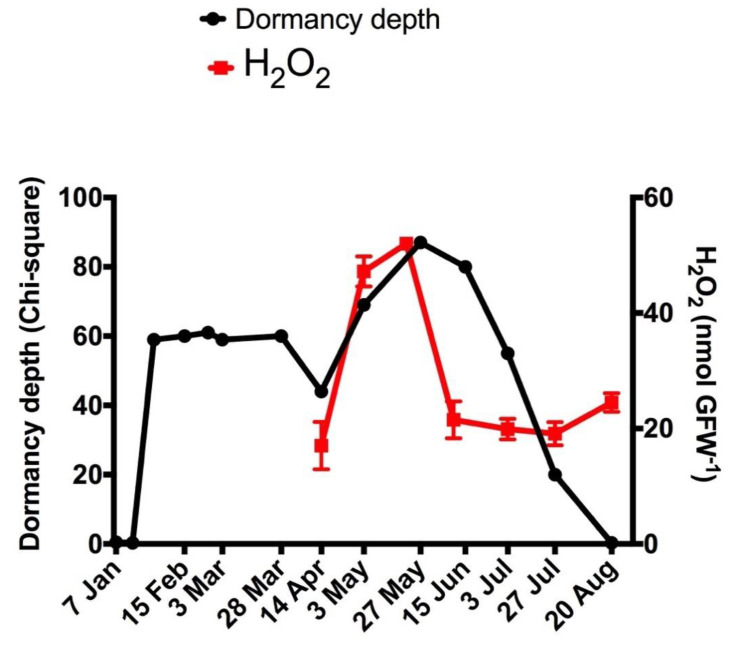
Relationship between the degree of endodormancy and H_2_O_2_ levels in Thompson seedless grapevine buds grown in Santiago, Chile. The degree of dormancy was determined by means of a log-rank test between the probabilistic function KM of the buds collected before (reference) and after the onset of the ED [26]. H_2_O_2_ was determined by a chemiluminescence method [29].

**Figure 2 antioxidants-10-00873-f002:**
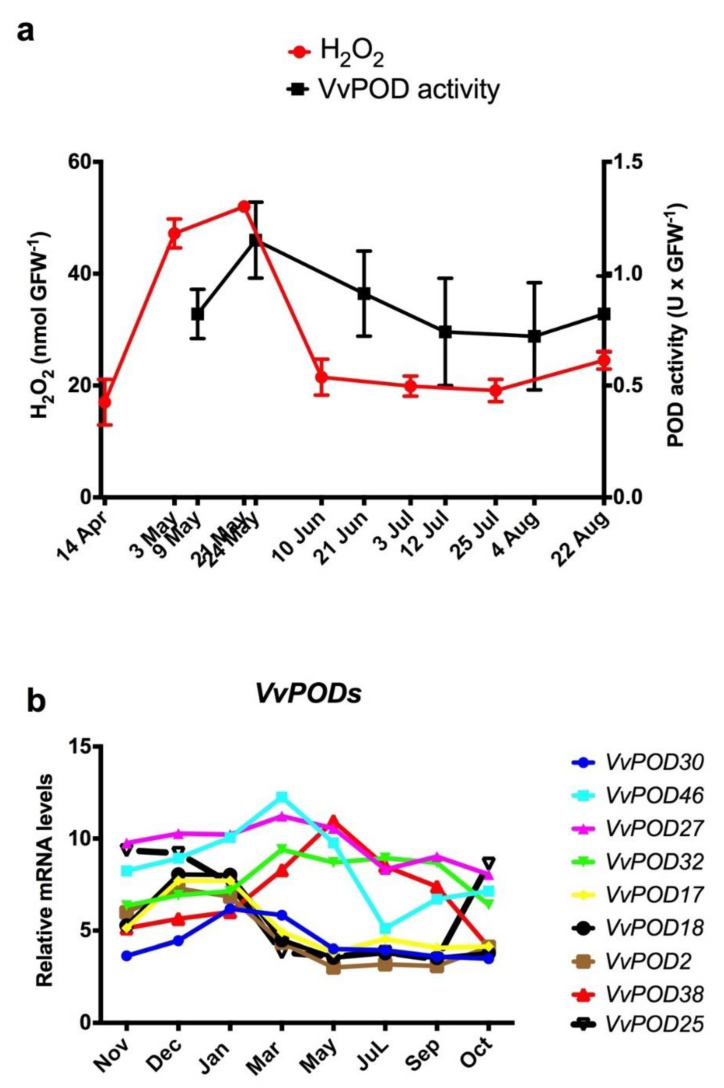
Expression of peroxidase genes (*VvPODs*), peroxidase activity and H_2_O_2_ levels during grape bud endodormancy. (**a**) Relationship between H_2_O_2_ levels and peroxidase activity (VvPOD) during endodormancy in Thompson seedless grapevine buds grown in Santiago, Chile. (**b**) Changes in the expression of peroxidase genes (*VvPOD*) throughout bud growth in Tempranillo grapevines grown in Alcalá de Henares, Madrid [21] and adapted to the Southern Hemisphere conditions.

**Figure 3 antioxidants-10-00873-f003:**
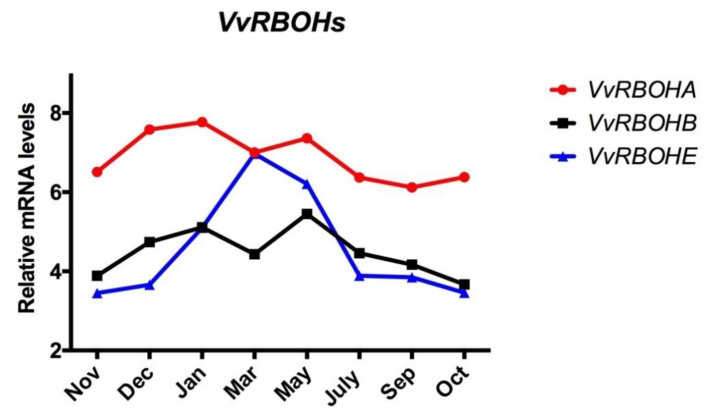
Expression profile of *RESPIRATORY BURST OXIDASE HOMOLOGS* (*VvRBOHs*) genes in grapevine buds during endodormancy. The expression profile of *VvRBOH* genes in Tempranillo grapevine buds grown in Alcalá de Henares, Madrid [21] and adapted to the Southern Hemisphere conditions.

**Figure 4 antioxidants-10-00873-f004:**
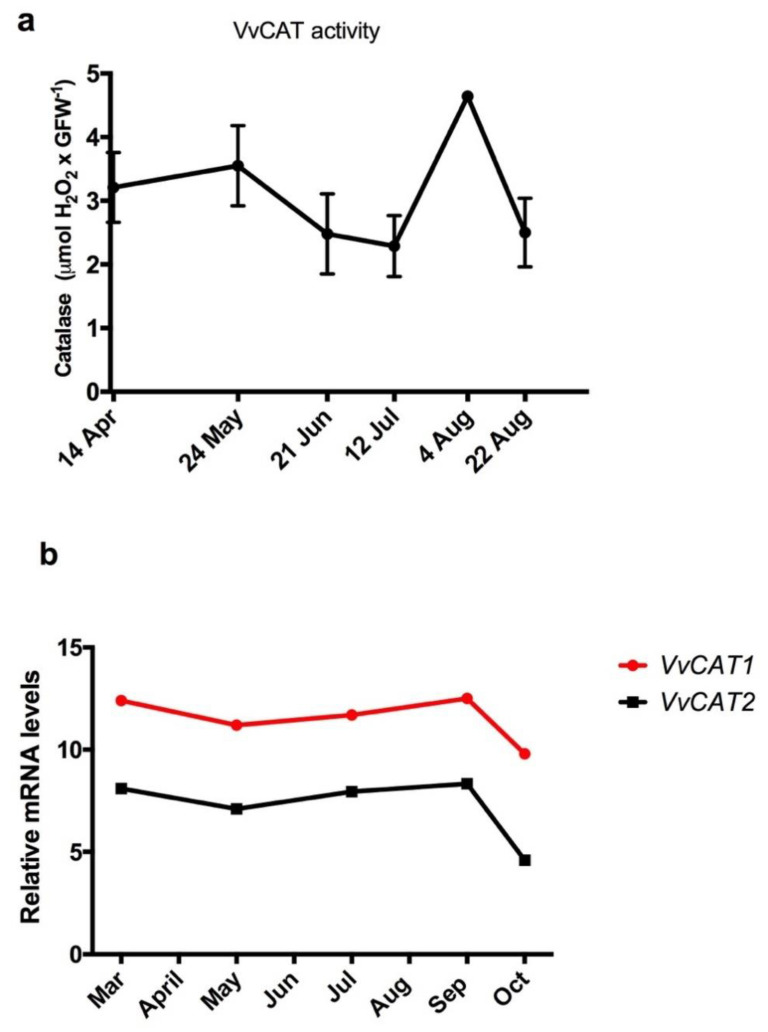
Catalase activity and expression of catalase genes in grapevine buds during endodormancy. (**a**) Catalase activity during endodormancy in Thompson seedless grapevine buds grown in Santiago, Chile. (**b**) Expression of catalase genes (*VvCAT1* and *VvCAT2*) in buds of Tempranillo grapevines grown in Alcalá de Henares, Madrid [21] and adapted to the Southern Hemisphere conditions.

**Figure 5 antioxidants-10-00873-f005:**
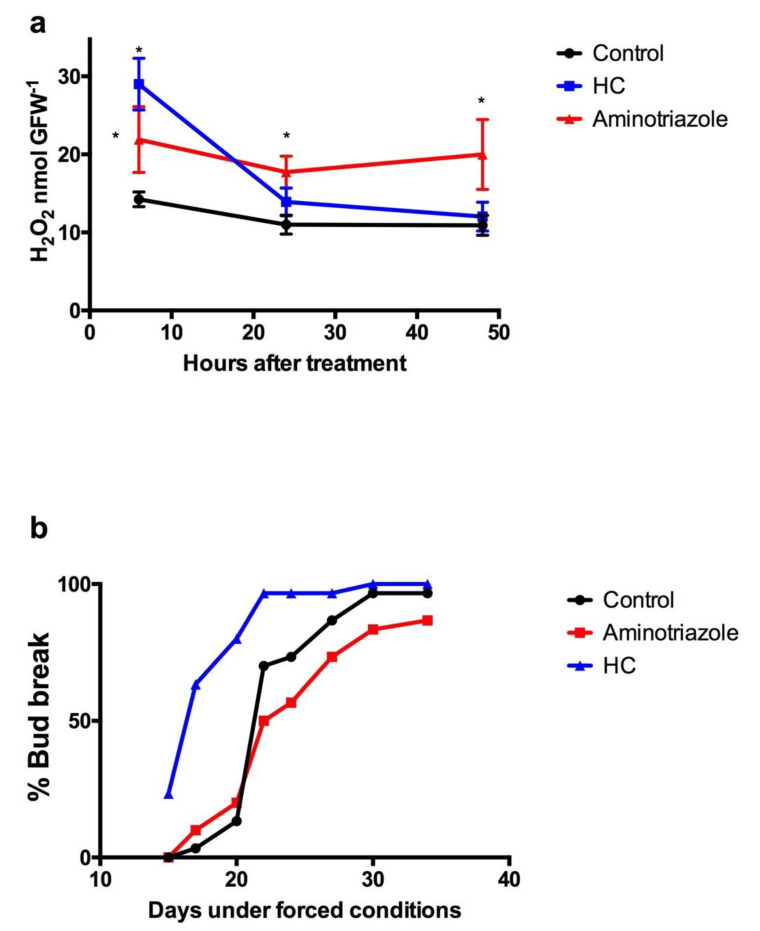
Aminotriazole effect on H_2_O_2_ content and budbreak in grapevine buds. The effect of aminotriazole (**a**) on the level of H_2_O_2_ and (**b**) on the budbreak of Thompson seedless grapevine buds was analysed. Buds collected at June 11 were excised into single-bud cuttings and sprayed with 2.5% HC, 2% Aminotriazole and water as control. The treated single-bud cuttings were placed in the growth chamber and H_2_O_2_ was determined at the desired time by a chemiluminescence method [29]. Values are the average of three biological replicates ± SD; and (*) indicates statistically significance differences (Dunnett’s multiple comparison test α = 0.05). Buds collected at July 18 were excised into single-bud cuttings (30 per treatment) and sprayed with 2.5% HC, 2% Aminotriazole and water as control. The treated single-bud cuttings were placed in the growth chamber and budbreak was assessed by the presence of green tips.

**Figure 6 antioxidants-10-00873-f006:**
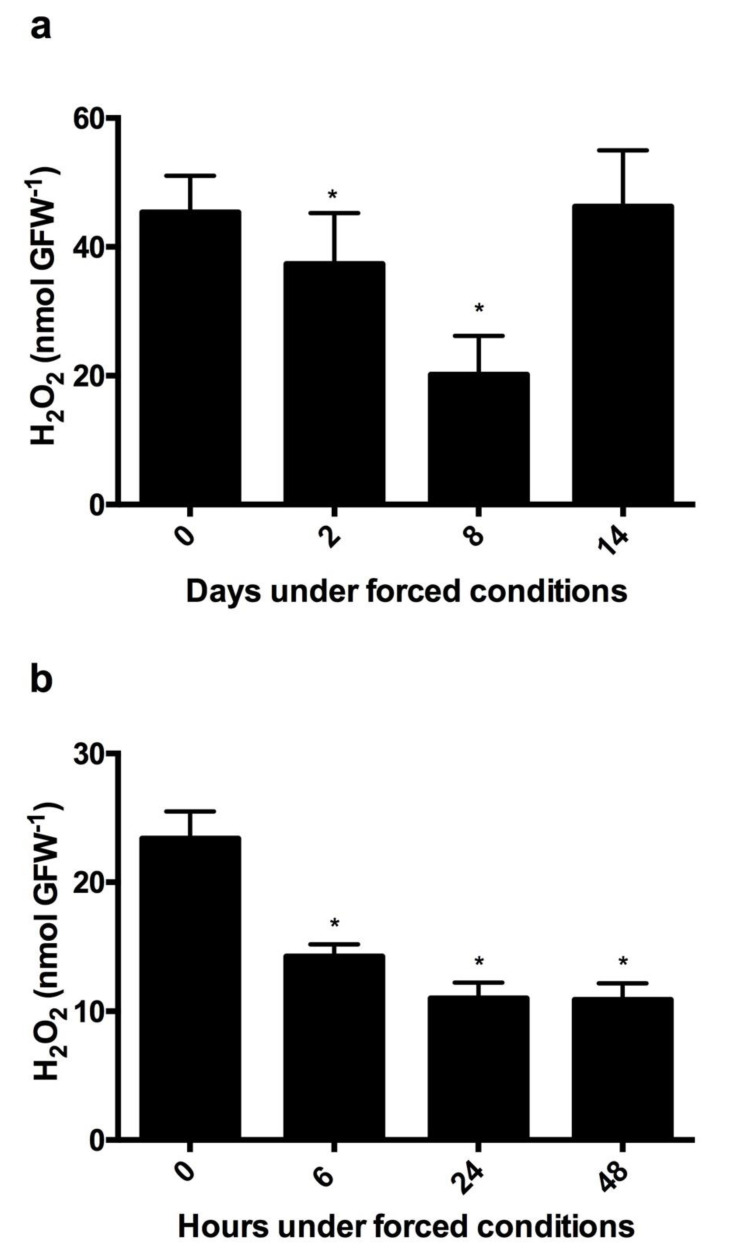
Rate of H_2_O_2_ reduction in dormant and quiescent grapevine buds under forcing budbreak. Decreases in H_2_O_2_ content in (**a**) endodormant and (**b**) quiescent buds of Thompson seedless grapevines under forced growth conditions. H_2_O_2_ was determined by a chemiluminescence method [29]. Values are the average of three biological replicates ± SD; (*) indicates statistically significance differences. Student’s test (α = 0.05).

**Figure 7 antioxidants-10-00873-f007:**
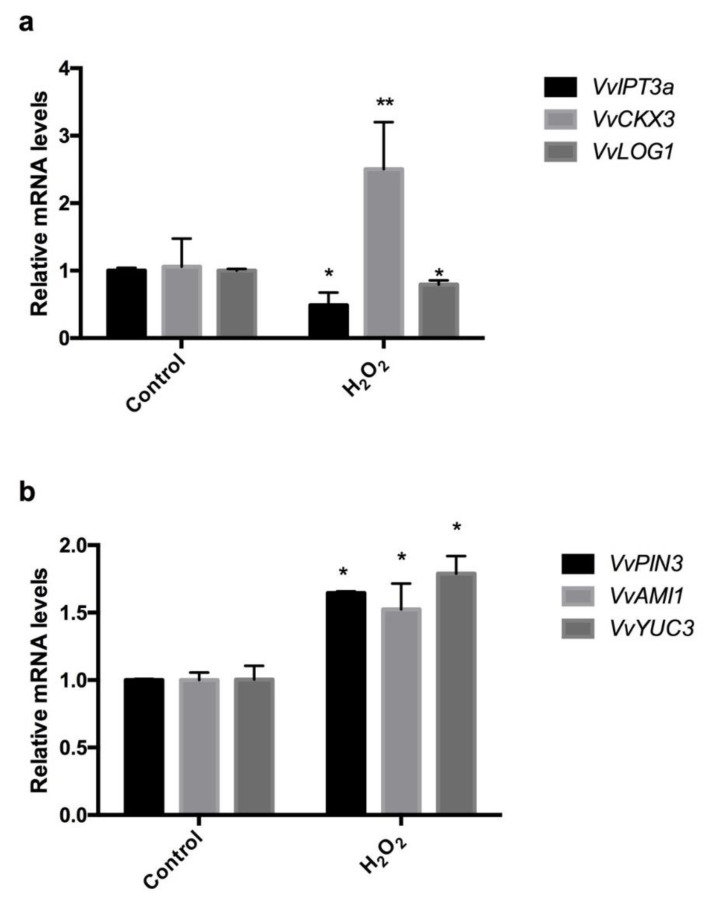
Transcriptional effect of H_2_O_2_ on the homeostasis of CK and auxin related genes. Effect of H_2_O_2_ applications on the expression of (**a**) cytokinin (CK) *VvIPT3a*, *VvLOG1*, and *VvCKX3* and (**b**) auxin *VvAMI1*, *VvYUC3* and *VvPIN3* genes. Gene expression analysis was performed by RT-qPCR and normalised against *VvUBIQITIN*. Values are the average of three biological replicates with three technical repetitions ± SD; (*) indicates statistically significance differences *p* ≤ 0.05; (**) *p* ≤ 0.01 (Dunnett’s multiple comparison test).

## Data Availability

Data is contained within the article.

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
