# Peer review of "Hydrogen Peroxide Increases during Endodormancy and Decreases during Budbreak in Grapevine (*Vitis vinifera* L.) Buds"

_antioxidants, 2021, doi:10.3390/antiox10060873_

Round 1
Reviewer 1 Report
The work of Pérez and colleagues characterizes the H202 content and the expression profiles of candidates enzymes during bud endodormancy and budbreak in grapevine.
They then connect their finding with other hormones as CK and IAA.
The work is clearly presented, and the methods well explained and nicely conducted. The work is interesting and provide additional information about the intricate relations among hormone and messengers molecules during endodormancy and budbreak.
I have some issues:
In figure 1A, authors detect a rapid increase in H202 from middle April, and this nicely correlates with the increase in dormancy depth. Later, they detect a sharp decrease in H202 content around the 15 of June, but the decrease of dormancy depth seems not to follow the H202 curve: the decrease in dormancy depth is not as rapid as the decrease in H202, and later again in August, H202 seems to start to pick again, but the dormancy depth goes to completely opposite direction. How the authors explain this completely opposite role of H202?
In figure 2a and b, authors check VvPOD activity, with the goal to associate its activity with H202 content. They talk about “POD activity”, which I believe can be misinterpreted. They detect a higher peroxidase activity is consequence of the higher POD genes level detected in May. So the higher activity is a reflection of the protein abundance in the tissue.
A similar comment is for the CAT enzymes in Figure 4A.
Related to Figure 2b, authors in the text says that there are 30 PODs genes expressed in the grapevine bud. Why did they choose to look at only nine of those in the 2b chart?
In figure 4A, it id difficult to compare the peaks among the two charts since they have a different X axis scale. Please plot the two sets of data to make them easily comparable.
I believe that figure 5b is not mentioned in the text. In the text, authors refers to a 14days assay, whereas in figure 5b it is reported a short time-course along 48 hours.
In Figure 5B, the level of H202 at 48h is about half of the value at 0hour time point. This is different in figure 5A, where at two days (=48hours), H202 content is only marginally reduced. Can authors speculate in what would be the different among these two experiment to originate these differences?
In the last part of their work the authors investigate the expression of some CK-related and IAA-related genes. It is not clear how the authors have chosen, among the vast families of factors involved in CK and IAA homeostasis, specifically VvIPT3, VvLOG1, VvCKX3, VvPIN3, VvAMI1 and VvYUC3. Please provide an explanations for the rationale behind your experimental strategy. This comment is valid also for the IAA-related genes.
By detecting decrease in transcripts abundance for VvLOG1 and VvIPT3, and higher expression of VvCKX3, authors conclude that CK content in the bud decreases upon H202 treatment. This can be of course the case, but the analyses provided by the authors do not directly demonstrated that, as only mRNA abundance is detect. If the authors want to conclude that CK levels are lower in the bud, they should quantify CK content by liquid chromatography for instance.
This is valid also for the auxin related genes. Detection of higher level of VvPIN3 transcripts does not directly correlate with increased auxin transport. Besides expression, PINs also changes their cellular polar distribution, which is the aspect that influences the most auxin fluxes and concentrations.
The results obtained for the auxin related genes are not mentioned in the discussion, so it would be useful have the authors commenting on that too.
I think that the last paragraph about the tole of aminotriazole is out of context at this position of the manuscript. If the authors want to present these data, they could include them in the paragraph about catalase activity.
Author Response
1.- We believe that endodormancy in grapevine buds is not driven by H2O2, and that changes in H2O2 level during ED development are the result of changes in ABA level. In addition, since the level of H2O2 can be affected by several factors, we postulate that during the development of ED, it is mainly controlled by the VvPODs genes, which in turn are regulated by ABA. At the end of the ED, ABA levels decrease, but another factor such as an increase in bud respiration, could be responsible for a further increase in H2O2 levels as seen in Figure 1A.
2.- Yes, we agree with the reviewer, and to avoid the misunderstanding, we incorporate a sentence clearing that point. Line 366-368 for VvPOD activity and Line 442-444 for VvCat activity
3.- We analysed only 9 of the 30 VvPODs genes described because these 9 genes are the ones that undergo changes in their expression during the development of ED, the others did not change their expression.
4.- Figure 4 was changed according to the reviewer suggestion
5.- Yes the reviewer is right, we rewritten all this paragraph Line 457-463
6.- The difference is explained in the text Line 461-463, and is due to the fact that the buds are in different developmental phase. When they are endodormant the drop in H2O2 is low, while when ED have finished and they are quiescent, the drop in H2O2 is rapid under condition of forced budbreak.
7.- We select VvIPT3a, VvLOG1 and VvCKX3 genes as representative of CK homeostasis because the expression of these genes changes greatly during the sprouting of grapevine buds under forced conditions and the same is valid for the IAA genes (Noriega and Pérez, 2017).
8.- Yes, the reviewer is right, we only reported changes in the abundance of genes related to CK and IAA homeostasis, and we did not measure CK or IAA concentrations. So we only suggest possible changes in their concentrations.
9.- Now, the auxin related genes are mentioned in the discussion. Line 593
- The last paragraph was transferred as suggest by the reviewer.
Reviewer 2 Report
Dear authors,
This is an outstanding paper - I really enjoyed reading the science reported on in this manuscript - great job - well done.
I have made a few comments and suggests throughout, simply to further increase the impact of your excellent study - please find the annotated PDF attached

Author Response
Many thanks to reviewer 2 for your glowing comments. All your suggestions were incorporated into the revised text

Round 2
Reviewer 1 Report
Perez and colleagues have answered to my comments. I am very satisfied with the new version of their work.
Just two small last things to check:
- the authors said the have modified the Figure 4 to present the two charts with the same X axis. However, the new version of the manuscript still contains the old versions of the charts.
- the authors followed my suggestion of moving the paragraph about aminotriazole to a different session; however that paragraph is still at the end of the results session in the revised version of the manuscript.
Author Response
1) We cannot modify figure 4a and figure 4b so that they present the same x-axis, this because they are the result of different experiments carried out by different research groups. The results of Fig. 4a were obtained by us, and the results of Fig. 4b were taken from the database of the article by Díaz-Riquelme et al. [18].
2) Now, we understand your point, and the aminotriazole paragraph was transfered to the result section in the catalase subsection Line 539-543.